# Modeling atypicality inferences in pragmatic reasoning

## Abstract

Empirical studies have demonstrated that when comprehenders are faced with informationally redundant utterances, they may make pragmatic inferences to accommodate the informationally redundant utterance (Kravtchenko and Demberg, 2015; Kravtchenko, 2021). Consider for instance the second utterance in *John went shopping. He paid the cashier.* As paying the cashier is easily inferable in the context of shopping, the utterance is redundant, and has been shown to raise an atypicality implicature, namely that John doesn't usually pay the cashier. We name these inferences triggered by a redundant utterance a *habituality* inference. Previous work has also shown that the strength of these inferences depends on prominence of the redundant utterance – if it is stressed prosodically, marked with an exclamation mark, or introduced with a discourse marker such as "Oh yeah", habituality inferences are stronger (Kravtchenko, 2021; Ryzhova and Demberg, 2020).

The goal of the present paper is to propose a computational model that can capture both the habituality inference and the effect of prominence. Using the rational speech act model, we show that habituality inferences can be captured by introducing joint reasoning about the habituality of events, similar to Degen et al. (2015); Goodman and Frank (2016). However, we find that joint reasoning models principally cannot account for the effect of differences in utterance prominence. This is because prominence markers do not contribute to the truth-conditional meaning. We then proceed to demonstrate that leveraging models which have previously been used to model low-level acoustic perception can successfully account for the empirically observed patterns of utterance prominence.

## 1 Introduction

When comprehenders encounter utterances that are pragmatically unexpected in the light of world knowledge, they may accommodate them by revising their beliefs about the common ground. Research on pragmatic inferences has to date paid relatively little attention to such common ground inferences, and formal models of pragmatic reasoning, with the notable exception of Degen et al. (2015), similarly do not typically account for the effects unexpected utterances may have on background beliefs about the world. As Degen et al. (2015) show, these may substantially alter utterance interpretation. In this paper, we present a Rational Speech Act (RSA) model (Frank and Goodman, 2012; Goodman and Stuhlmüller, 2013) of how background beliefs about activity habituality may be updated upon encountering informationally redundant activity descriptions. The utterances we concern ourselves with are the following:

1. "John went shopping."

2. "John went shopping. *He paid the cashier.*"

3. "John went shopping. *He paid the cashier!*"

4. "John went shopping. ***Oh yeah, and*** *he paid the cashier.*"

In each case, the speaker first establishes that a stereotypical series of actions, such as the *shopping* event, occurred. In the case of utterance (1), the speaker then stops. However, given world knowledge about the structure and typical activity components of such events, most listeners conclude that an activity as habitual as *paying the cashier* must have taken place, even if not mentioned explicitly (Bower et al., 1979). This raises the question of what interpretation, exactly, a listener may assign to a redundant utterance, such as that in (2-4). We assume that listeners expect for rational speakers to not be unnecessarily verbose and to hence omit information that does not need to be explicitly stated to be inferred accurately.

Kravtchenko and Demberg (2015) have established, using a variety of experimental materials

that follow the pattern above, that upon reading an utterance that is redundant at face value, as in (2-4), the listener infers that *John* must be a less habitual payer than initially assumed, as this is one of the few ways to justify the activity's explicit mention. Figure 1 shows the results from their study, which shows listener distribution of *habituality* estimates after reading each of the above utterances. These estimates were obtained by asking participants to provide ratings of how habitual a given activity was, in the context of a particular activity sequence, see the section on the habituality prior for more details.

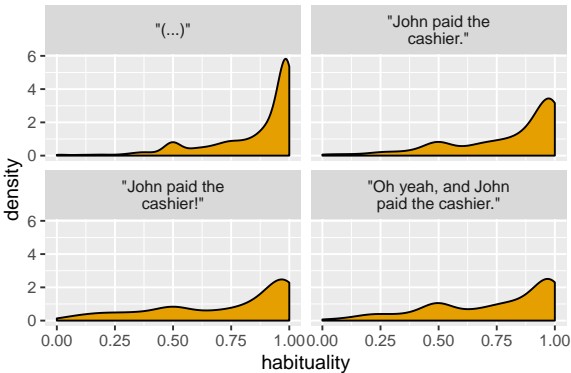

Figure 1: Smoothed distributions (kernel density estimates) showing the relative strength of *habituality* inferences following different utterances. Stronger inferences are reflected by fewer *habituality* estimates at the right end of the distribution.

In Figure 1, it can be observed that *habituality* estimates – i.e., how often *John* is expected to *pay the cashier* – are overall rather high in the *null utterance* condition, as would be expected. However they noticeably (and significantly) lower when encountering a *plain* redundant utterance, which ends only in a period. They further (and significantly) decrease when the utterance is either followed by an exclamation mark, which denotes implicit prosodic emphasis (the *exclamation* condition); or when there is a discourse marker on the utterance signaling the utterance's relevance to the discourse or listener (the *oh yeah* condition).

In this paper, we therefore aim to also model the interpretation of truth-conditionally equivalent utterances that differ in their perceptual prominence. Longer or more prominent utterances can be thought of as being perceived and attended to better than less prominent but meaning-equivalent utterances. More prominent or attention-drawing utterances should therefore strengthen pragmatic inferences (cf. Wilson and Sperber, 2004). It turns out, however, that a base RSA model principally cannot predict inferences of different strengths for the different utterance prominence conditions, as will be discussed more in the section on *Attentional prominence and inference strength*.

The issue of attentional prominence is related to work on the pragmatic interpretation of prosodic stress and the comprehension of utterance fragments (Bergen and Goodman, 2015). They model effects of prosodic stress on utterance interpretation in terms of a noisy-channel model which is incorporated into the standard RSA model. In the present paper, we explore whether this solution can also predict the qualitative findings on habituality inferences. Conceptually, this means that we propose to extend the noisy channel RSA model from issues related to low-level perception of prosody or short utterance fragments to longer multi-word utterances. While our target utterances are unlikely to be fundamentally misheard or misperceived at a low level, they are more or less likely to be remembered or recalled accurately, based on whether they drew the listener's attention. As we show, the noisy channel machinery introduced by Bergen and Goodman (2015) can be generalized to a communication channel where varying degrees of utterance prominence can result in more or less accurate recall, with very promising results.

To summarize our criteria for a successful model, the predicted, and in the case of (a) and (b), empirically validated (Kravtchenko and Demberg, 2015), effects associated with the use and comprehension of such utterances are:

a) As *paid the cashier* in (2-4) is informationally redundant, this utterance at face value is pragmatically odd. Listeners resolve this by assuming that cashier-paying is *not*, in fact, typical for this individual and in this context, contrary to their prior beliefs.

b) Expending more articulatory effort on an informationally redundant utterance, for example by way of using exclamatory prosody, should strengthen the inference, as increased articulatory effort reflects greater speaker intent to transmit precisely this message to the listener.

c) Speakers should preferentially use more attentionally prominent utterances to transmit

**Q: How often do you think John usually pays the cashier, when grocery shopping?**

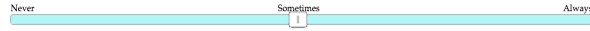

Figure 2: This is a slider, as used in the experiment.

## 2 The habituality prior

(Kravtchenko, 2021) collected ratings for the habituality of various activities, from 2100 participants. These participants were asked to indicate, on a sliding scale from never to always, how often they thought someone engaged in a particular activity (such as paying the cashier) when engaged in a certain event sequence (such as going shopping), see Figure 2 for an example. The slider was discretized into numbers 0 to 100 for analysis of the data.

Importantly, the question was asked in the context of a story which introduced the script, but did not contain the target utterance, similar to condition (1) in the example in the introduction. These ratings are used in our study to estimate the habituality prior. Figure 3 shows the distribution of *habituality* estimates collected from participants. We can see that most participants believe that the event is highly likely to have happen, as visible from the high number of ratings in the upper quarter of the slides. Note that there is also a number of ratings around 50%, with a dip in ratings round the 50% mark. We believe that this bimodality with a peak around 50% is an artifact related to the method of data collection, reflecting the well-known midpoint bias. This midpoint bias affects all of the conditions. However, for our models, we do not aim to replicate this midpoint bias, as the RSA model is not intended to capture the process of how beliefs are expressed along a scale. Rather, we aim to model the overall pattern of the distribution, i.e. the proportion of responses qualifying the event as highly likely vs. the heavier tail.

A beta distribution was fit to the distribution of responses using the `fitdistrplus` R package (Delignette-Muller and Dutang, 2015; R Core Team, 2018), and fed directly into the model.

## 3 Modeling a shift in background beliefs

The literal meaning of *paid the cashier* communicates nothing about activity *habituality* directly. Standard RSA models, where the listener only in-

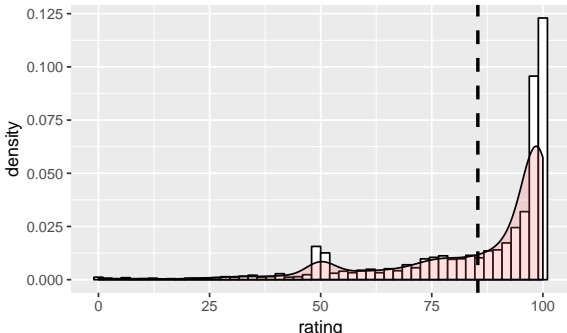

Figure 3: This figure shows the empirical prior estimate for habitual utterances. The dotted line represents the mean. The smoothed distribution is a kernel density estimate.

fers a world state given an utterance, can therefore accurately predict only that the *cashier* was definitely paid in the case of utterances (2-4), and that they may or may not have been paid, modulated by prior beliefs about the habituality of *paying*, in the case of utterance (1). Activity habituality by itself cannot be modeled within this framework, since all utterances are at face value equally consistent with all possible *habitualities*.

To model the *habituality* inference, it is necessary to minimally incorporate joint reasoning about background knowledge, very similarly to Degen et al. (2015). Here, the listener reasons jointly about the current world state ($s$) (i.e., did the activity in question occur, or not), as well as the true *habituality* of the activity ($h$), given the speaker's utterance ($u$). The relative weights that speakers give to the cost and utility functions are represented by $\lambda$, in the case of the speaker's prioritization of reducing utterance cost; and $\alpha$, in the case of the speaker's maximization of utterance utility[1]. $\lambda$ is set at 1, and $\alpha$ is set at 7, to optimize the quantitative fit with empirical results. Only one level of recursion is used, as is standard, given limited empirical evidence for deeper levels of recursion in online pragmatic reasoning (Goodman and Stuhlmüller, 2013; Goodman and Frank, 2016).

### 3.1 hRSA model

A standard RSA model which incorporates joint reasoning (e.g., Degen et al., 2015; Goodman and Frank, 2016) can model both changes in beliefs about the world, and changes in beliefs about the

---

[1]See https://michael-franke.github.io/probLang/chapters/app-03-costs.html, for a discussion of how utterance costs $C$ should be formalized within the RSA framework.

current activity state. Here, we feed our empirical priors directly into the model, where the likelihood of the activity occurring is conditional on the activity *habituality*. Whether a given activity occurred, or not ($s$), then, is simply a Bernoulli trial with $p = h$. In the hRSA model, the literal listener arrives at the most likely current world state ($s$) (whether the activity took place, or not), given the utterance ($u$), and prior beliefs about activity habituality ($h$):

$$P_{L_0}(s|u, h) \propto [\![u]\!](s) \cdot P(s|h)$$

$L_0$ does not reason about *habituality*, as this is not a part of the literal interpretation. The pragmatic speaker, $S_1$, considers the likelihood that a given utterance will communicate the current activity state to the listener, given common-ground beliefs about *habituality*, while balancing the cost $C^1$ of uttering the potential utterances relative to one another:

$$P_{S_1}(u|s, h; \alpha, \lambda, C) \propto$$
$$P(u; \lambda, C) \exp(\alpha \log P_{L_0}(s|u, h))$$

The pragmatic listener, $L_1$, considers the likelihood that a given utterance would be chosen by the speaker, given the probabilities of particular world states and activity habitualities, and arrives at the most likely interpretation of the utterance on this basis:

$$P_{L_1}(s, h|u) \propto P_{S_1}(u|s, h; \alpha, \lambda, C) \cdot P(s|h) \cdot P(h)$$

Note that in the present hRSA model, the habituality is a type of belief about the world, like in other joint reasoning models. However, it is not a belief about the habituality of an activity with respect to the speaker themself, or about the habituality of the activity in general. Instead, the habituality inference is intended to capture whether the general habituality of an activity generalizes also to the agent of the story (Generally, people pay when going shopping, hence John pays when going shopping), or whether the habituality of the activity does not extend to John, i.e., *John* doesn't usually pay.

### 3.2 Results

The hRSA model correctly captures predicted effect (a), as seen in Figure 5: if an activity is described explicitly, the *habituality* is likely to be low. What it does not, however, capture is that there is no possibility of simply leveraging utterance costs to capture effects (b) and (c) above, as seen in Figures 4 and 5, respectively.

Let us first consider the model predictions for the speaker (Figure 4): An example of an activity with 95% habituality could be paying the cashier, as we can see, the speaker would be predicted not to mention this explicitly. An activity with 50% habituality could for instance be buying apples. Here we can see that the speaker would be predicted to prefer a plain utterance, with some probability also distributed among the other choices. For a surprising event with just 5% habituality, which could for instance correspond to accidentally dropping something, the speaker's utterance choices are almost identical to the ones for 50% habituality. The only difference is that the empty utterance is not predicted. In particular, speakers are very reluctant to use exclamation marks or other markers even in this condition. To conclude, speaker utterance choice according to the hRSA model is not fully consistent with hypothesis (c).

Furthermore, we can see in Figure 5 that there is virtually no effect of utterance prominence on interpretation by the pragmatic listener. This is at odds with the empirical data and hypothesis (b).

The crucial point for understanding this failure to show the desired effects is as follows: There are three possible ways of articulating the redundant utterance: with a full stop at the end (2); with exclamatory prosody (3); or with an attention-drawing and relevance-establishing discourse marker (4), in order of increasing utterance cost. The more attentionally prominent utterances (3-4) will never be of any advantage to the literal listener, in terms of whether they effectively communicate the current world state. They are likewise of no advantage to the speaker, either in terms of likelihood of accurate message transmission to the literal listener, or the speaker's presumed goal to conserve articulatory effort. As a consequence, the pragmatic listener will not infer that the more effortful utterance is more likely to communicate an atypical meaning.

The failure of standard RSA models to derive pragmatic inferences of different strengths, given semantically meaning-equivalent utterances, is directly analogous to their failure to derive M-implicatures or inferences due to prosodic stress, as detailed and mathematically proven in Bergen et al. (2016). We consider this a serious failure of the base RSA model. As a result, this framework fails to model any of the empirically demonstrated

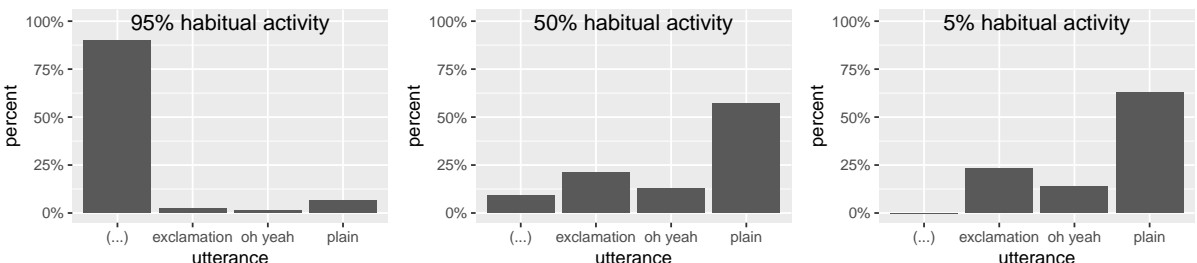

Figure 4: This figure shows the speaker's utterance preferences in the hRSA model.

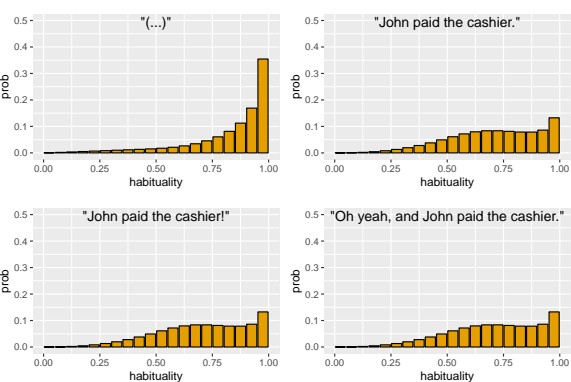

Figure 5: This figure shows the relevant posterior (pragmatic listener) measures for the hRSA model.

Table 1: Confusion matrix showing the likelihood of any given utterance being perceived as any other.

|  | (…) | He paid. | He paid! | Oh yeah… |
|---|---|---|---|---|
| (…) | **0.99** | 0.01 | 0.0001 | 0.0001 |
| He paid. | 0.01 | **0.95** | 0.02 | 0.02 |
| He paid! | 0.0001 | 0.02 | **0.97** | 0.01 |
| Oh yeah… | 0.0001 | 0.02 | 0.01 | **0.97** |

effects that increased utterance salience has on utterance choice or comprehension, also predicted by psycholinguistic theories of language comprehension (e.g., Levy, 2008).

## 4 Attentional prominence and inference strength

In order to capture effects (b) and (c), it is necessary to assign some attentional or memory-related benefit to the more costly redundant utterance, to be already active at the $L_0$ level. Empirically, there is evidence that readers often cannot recall whether elements in a stereotyped activity sequence were explicitly mentioned, or not (Bower et al., 1979), and that informational redundancy, even at the multi-word level, in part serves the purpose of ensuring that listeners attend to and accurately recall relevant information (Walker, 1993; Baker et al., 2008). The noisy-channel RSA model proposed by Bergen and Goodman (2015), with minimal modification, successfully captures this intuition, although in our case we consider the probability that an utterance is attended to and stored in memory, rather than simply misheard, as represented in Table 1.

The exact values in the table are set somewhat arbitrarily, was we do not have empirical data about this. The most important aspect to capture the empirical effects is that the utterances that are prominent and attract more attention should have a very small confusion probability with the null utterance, in particular much smaller than the probability of the full stop condition.

We chose the values shown in the table as we found them to be intuitively plausible. On the diagonal, that each utterance is most likely to be recalled and remembered as itself. Each utterance also has a small likelihood of being misperceived as a perceptually 'neighboring' utterance: *"he paid!"* and *"oh yeah..."* both have a small likelihood of being mistakenly recalled as the other, and a higher likelihood of being recalled as the plain utterance: *"he paid."* The plain utterance ("*he paid.*"), which does not draw any particular attention, has a small likelihood of being remembered or recalled as nothing ("*(...)*"), and the 'null' utterance ("*(...)*") may be mistakenly recalled as the plain utterance. Although this last confusion may appear counterintuitive, (Bower et al., 1979) shows that in script contexts, participants frequently recall reading about habitual activities that were not, in fact, mentioned explicitly. To sum up the general intuition, the listener is more likely to notice and accurately recall more perceptually prominent utterances, and is correspondingly more likely to wonder why the speaker went to the extra effort in producing these utterances.

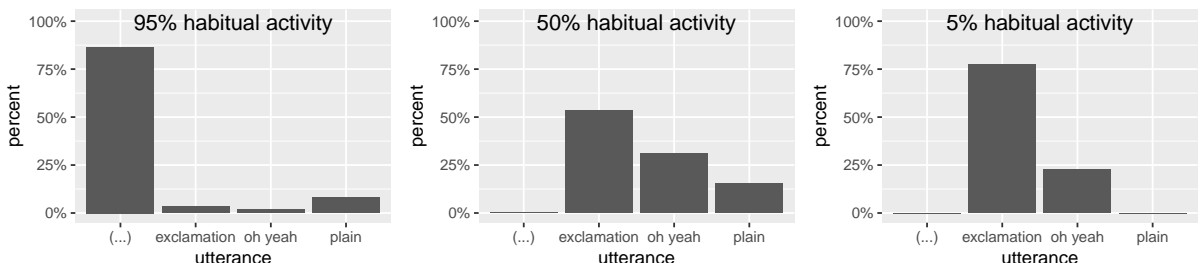

Figure 6: This figure shows the speaker's utterance preferences in the noisy channel hRSA model.

### 4.1 Noisy hRSA model

In the noisy channel hRSA model, it's assumed that every utterance has a non-trivial likelihood of not being actively attended to, and being mistaken for or mis-recalled as a 'neighboring' utterance. Here, $u_i$ represents the utterance intended by the speaker, and $u_r$ represents the utterance actually recalled by the listener. At every level, the listener or speaker reason about the likelihood that the utterance they actually perceived is not the utterance that was uttered, or, conversely, that the utterance they intend may not be the utterance that is in fact perceived. Again, only one level of recursion is used, and only one is necessary to capture these results. To note, given the mathematical properties of RSA models (Bergen et al., 2016), deeper levels of recursion would not in themselves alter the ability of the model to capture the utterance-dependent variations in inference strength.

The literal listener reasons about the likely world state given the utterance they in fact recall, and the habituality of the activity in question. However, they weight this by the likelihood that the utterance recalled is not in fact the utterance that was intended.

$$P_{L_0}(s|u_r,h) \propto$$
$$[\![u_r]\!](s) \cdot P(s|h) \cdot \sum_{u_i:[\![u_i]\!](s)=1} P(u_r|u_i)P(u_i)$$

The pragmatic speaker chooses an utterance $u_i$ given a world state and activity *habituality*, taking into consideration the likelihood that the listener may misremember or mis-recall the utterance they intend. Intuitively, this is far more likely to have consequences when the meaning the speaker intends to transmit is unexpected by the listener.

$$P_{S_1}(u_i|s,h;\alpha,\lambda,C) \propto$$
$$P(u_i;\lambda,C)\exp(\alpha \sum_{u_r} P(u_r|u_i)\log P_{L_0}(s|u_r,h))$$

The pragmatic listener, as in the hRSA model, infers the current world state and activity habituality, taking into account the conditions under which the speaker made their utterance choice, their habituality prior, and the likelihood of the state given the habituality. They again take into account the possibility that they may mis-recall the speaker's intended utterance.

$$P_{L_1}(s,h|u_r) \propto P(s|h) \cdot P(h) \cdot$$
$$\sum_{u_i} P_{S_1}(u_i|s,h;\alpha,\lambda,C)P(u_r|u_i)P(u_i)$$

In sum, a redundant event description that does not somehow draw the listener's attention is less likely to be attended to, and more likely to be misperceived or misremembered by the literal listener as a '*null*' utterance. The pragmatic speaker must take into account that their utterance may not be attended to or remembered by the listener, and the pragmatic listener likewise considers the possibility that they may fail to attend to or remember what the speaker uttered.

### 4.2 Results

For high-habituality activities, as in the plain hRSA model, speakers are very unlikely to describe the activity explicitly – and if they do, they tend to choose less effortful utterances, as shown in Figure 6. Moderately habitual activities are only moderately likely to be mentioned, and again speakers gravitate towards less effortful utterances. The strength of this effect can be modulated by changing alpha and/or by changing the confusion matrix. The result pattern is consistent with expectations, as moderately predictable activities are less likely to be assumed to have not occurred – it is therefore not quite as important to grab the listener's attention to ensure that they do, in fact, understand that the activity took place. Non-habitual activities are virtually always described explicitly, and as can be seen, speakers prefer a higher-effort utterance that is more likely to be attended to, and less likely to be mis-recalled as a "*null*" utterance. This matches our predicted effect (c).

This model qualitatively captures all three pre-

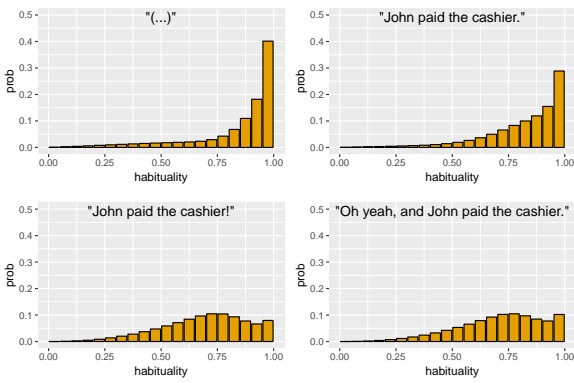

Figure 7: This figure shows the relevant posterior (pragmatic listener) measures for the noisy channel hRSA model.

dictions set out above. As can be seen in Figure 7, pragmatic listeners adjust the common ground such that a typical activity which is uttered overtly is inferred to be less habitual: the peak at the high ratings is greatly reduced in favour of a heavier tail. This is thus reflective of hypothesis (a). The figure also shows that more effortful utterances (3) and (4) lead to stronger *habituality* inferences (for high-*habituality* activities): the tails for the exclamation mark condition and the "oh yeah" condition are substantially heavier than for the full stop condition. This is in line with the empirical data in Figure 1 and with hypothesis (b) above. The heaviness of the tail depends on the settings of the alpha parameter: heavier tails without bimodality can be achieved by choosing lower values for alpha, thinner tails with a second peak between 0.6 and 0.7 can be achieved by choosing higher values for alpha. Furthermore, as shown in Figure 6, speakers are likely to use more effortful utterances to communicate less likely meanings, in line with hypothesis (c).

Comparing the empirical posterior habituality estimates from Figure 1 to the habitualities inferred by the pragmatic listener from Figure 7, we can see that the overall pattern is qualitatively similar.

## 5   Conclusion

Overall, the hRSA model shows that RSA models, incorporating joint reasoning about world knowledge, successfully model common ground habituality inferences, including in the case of "zero" utterances. Further, the noisy hRSA model extends the noisy-channel model in (Bergen and Goodman, 2015) beyond the level of relatively low-level

utterance perception, to higher-level attentional processes, which should generalize to other cases where truth-conditionally equivalent utterances differ in prominence and do not receive the same pragmatic interpretation. The noisy hRSA model demonstrates that the empirically observed effects can be captured well qualitatively, and that this can be accomplished using existing and independently motivated mechanisms, and does not require the postulation of any new mechanisms that fail to generalize beyond the phenomenon in question.

Limitations of our work include the fact that we did not empirically estimate the confusion matrix for utterances for the confusion matrix of the noise model, and consequently did not attempt an quantitative model fit for the noisy hRSA listener model. A further area of future work consists of empirically estimating to what extent speakers choose more prominent utterances as a function of how non-predictable/surprising the target utterance is.

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
