# OpenReview forum: "Modelling atypicality inferences in pragmatic reasoning"
_aclweb.org/ACL/2022/Workshop/CMCL — Submitted to CMCL 2022_

### Official Review · Reviewer_px8X · 2022-03-15
**Interesting study, could use some more attention to existing understanding of implicature and common ground**

**Rating:** 6
**Confidence:** 3

**Review:**

I'm not sure that the way the authors measured the habituality prior will really get accurate measurements. If we assume that paying the cashier logically follows from going shopping, then even asking the question of how often he pays the cashier may introduce the same sort of Q-implicatures that stating "he paid the cashier" does -- respondents might think "why ask me that unless there is reason to suspect that he often doesn't"? Thus, these ratings might not be giving a baseline habituality level; they might just be showing the same inferences that the main experiment is showing.

It seems to me that many potential implicatures may become available in the sorts of utterances used here. "John went shopping -- oh yeah, and he paid the cashier" might implicate that he doesn't usually pay, but it might implicate lots of other things as well, and it's not even clear to me that "John usually doesn't pay" is even the most likely implicature to arise (hence the rather small effect seen in the author's previous studies -- even in the more marked conditions, there is still a plurality of participants thinking that John usually pays). It's especially hard to know what implicature to recover here without any context (so it would be useful to see the full stories that were used in the study; if participants only see these utterances in a minimal context I suspect that their interpretations would be so all over the place there would be nothing interesting to model; the authors mention that these were put in story contexts but I don't see the full texts anywhere). It seems like it's easy for an interlocutor to notice that the utterance is weird (flouting the maxim of quantity, the Q-principle, or whatever we want to call it) and that the speaker intends to produce some implicature, but exactly what that implicature is is very underdetermined -- indeed this has been one of the core criticisms of Gricean pragmatics ever since it started.

The authors state near the beginning of the paper that there has been little research attention to pragmatic inferences that involve changing one's beliefs about the common ground, but this doesn't seem right. Quite a lot of theorizing in pragmatics and discourse analysis has assumed that communication is all about updating the common ground and mutual knowledge; e.g., contra stuff like Grice's theory of non-natural meaning, some approaches argue that the whole point of communication is to get an interlocutor to introduce something new into their understanding of the common ground. (e.g. Discourse Representation Theory.) Some pragmatic phenomena, particularly presupposition and conventional implicature, have been explained with recourse to how they put things into the common ground (e.g. work by Stalnaker, Potts; for a summary of work from the very dawn of pragmatics see e.g. chapter 4.4 of Levinson 1983).

---

### Official Review · Reviewer_ACau · 2022-03-22
**Interesting but too preliminary, with clear empirical limitations**

**Rating:** 4
**Confidence:** 4

**Review:**

This paper presents a formal model — within the RSA framework — of pragmatic inferences about the frequency of an activity that arise when utterances contain information that is in principle redundant (under default assumptions for an event type). The authors first present a version of the model (hRSA) that incorporates prior expectations given common ground knowledge, building on Degen et al (2015).  This model fails to account for the possible preference of speakers to use more costly utterances to convey unusual meanings and the fact that these more costly utterances strengthen the inference. To remedy this, the authors proposed a modified version of the model (noisy hRSA), which takes into account the memorability of a given utterance form, building on Bergen and Goodman (2015).

I have read the paper with interest, but do not think that it is suitable for CMCL. The paper makes theoretical predictions, which is a strength. However, the work in its current state is largely incomplete:

- the model revolves around one single example (the one given in the introduction) and no details about data are given (e.g., the data collected by Kravtchenko & Demberg (2015) is not described; nothing is said about how many types of events were targeted in the experiments by Kravtchenko (2021) regarding habituality priors);
- most importantly, no empirical evidence is provided for assumption (c), i.e., the hypothesized speaker behaviour which the model captures has not been empirically validated and it’s not clear why this should hold;
- similarly, no evidence for the claim that “speakers are very reluctant to use exclamation marks or other markers even in this condition” (lines 304-306);
- the values in table 1 are made up (“intuitively plausible” with no empirical evidence and not substantiated by a well-defined theory);

The authors acknowledge these weaknesses at the end of the conclusions. Given these limitations, my assessment is that the paper does not currently contain enough substance and is not a good fit for this workshop in particular.

Other comments:

- The RSA framework is given for granted. The paper should include a brief description of its main components. Space is clearly not an issue.

- The term “atypicality inferences” used in the title is never used in the paper, where instead the phenomenon studied is referred to as “habituality inferences”.

- What does hRSA stand for? (Is it “habituality RSA?). This acronym is used in section 3.1 without spelling out what it stands for, and without a citation (in line 246, it appears that this is not a contribution of the present paper but rather a model introduced in previous work).

- Lines 169 and 385: wrong citation format.

- Consider shortening the abstract, using one single paragraph, which is the established scholarly practice in the field.

---

### Official Review · Reviewer_HsPv · 2022-03-25
**Preliminary blend of two RSA extensions**

**Rating:** 6
**Confidence:** 4

**Review:**

The authors study "habituality" inferences using an example from a previously-collected dataset by Kravtchenko & Denberg, 2015). They attempt to derive these under the rational speech acts framework and find that a base RSA agent fails, but that a blend of habitual and noisy channel RSA variants can produce these inferences and can mirror the empirical observation that when the triggering statement is more prominent, the inference gets stronger.

This is an interesting but somewhat incremental extension to RSA theory, picking up on the Bergen and Goodman (2015) noisy channel RSA model. The theoretical contribution is to work out how these two model variants can be combined, and that seems quite appropriate for the current venue. Unfortunately, the empirical side is not yet well fleshed-out: the prior data are not described in any depth, and the evaluation is completely qualitative. Further, the model depends on establishing a confusion matrix for utterances that is (as noted as a limitation by the authors) currently simply invented. A minimum step here would be to conduct some kind of sensitivity analysis to show that it is not the specific details of this confusion matrix that produce the result.

In sum, this feels like a good start but some further evaluation is needed to cement the contribution of the work.

---

### Decision · Program_Chairs · 2022-03-29

Reject